

**Effects of ozone-vegetation coupling on surface ozone air**
**quality via biogeochemical and meteorological feedbacks**
Mehliyar Sadiq[1], Amos P. K. Tai[1,2], Danica Lombardozzi[3], and Maria Val Martin[4]
[1]Graduate Division of Earth and Atmospheric Sciences, Faculty of Science, Chinese University of Hong Kong,
Hong Kong
[2]Earth System Science Programme, Faculty of Science, Chinese University of Hong Kong, Hong Kong
[3]National Center for Atmospheric Research, Boulder, Colorado, USA
[4]Department of Chemical and Biological Engineering, University of Sheffield, Sheffield, UK
*Correspondence to*: Amos P. K. Tai (amostai@cuhk.edu.hk)
**Abstract.** Tropospheric ozone is one of the most hazardous air pollutants as it harms both human health and
plant productivity. Foliage uptake of ozone via dry deposition damages photosynthesis and causes stomatal
closure. These foliage changes could lead to a cascade of biogeochemical and biogeophysical effects that not
only modulate the carbon cycle, regional hydrometeorology and climate, but also cause feedbacks onto surface
ozone concentration itself. In this study, we implement a semi-empirical parameterization of ozone damage on
vegetation in the Community Earth System Model to enable online ozone-vegetation coupling, so that for the
first time ecosystem structure and ozone concentration can coevolve in fully coupled land-atmosphere
simulations. With ozone-vegetation coupling, present-day surface ozone is simulated to be higher by up to 6
ppbv over Europe, North America and China. Reduced dry deposition velocity following ozone damage
contributes to ~40-100% of those increases, constituting a significant positive biogeochemical feedback on
ozone air quality. Enhanced biogenic isoprene emission is found to contribute to most of the remaining
increases, and is driven mainly by higher vegetation temperature that results from lower transpiration rate. This
isoprene-driven pathway represents an indirect, positive meteorological feedback. The reduction in both dry
deposition and transpiration is mostly associated with reduced stomatal conductance following ozone damage,
whereas the modification of photosynthesis and further changes in ecosystem productivity (which are significant
per se) are found to play a smaller role in contributing to the ozone-vegetation feedbacks. Our results highlight
the need to consider two-way ozone-vegetation coupling in Earth system models to derive a more complete
understanding and yield more reliable future predictions of ozone air quality.

**1 Introduction**
Tropospheric ozone is one of the air pollutants of the greatest concern due to its significant harm to
human respiratory health. Increases of ozone since the preindustrial time have been associated with a global
annual burden of 0.7±0.3 million respiratory mortalities (Anenberg et al., 2010). Decades of observational
records have also demonstrated the damaging effect of surface ozone on vegetation and crop productivity
(Ainsworth et al., 2012). The phytotoxicity of ozone is shown to induce stomatal closure and reduce primary
production, with ramifications for climate through the modification of surface energy and water fluxes and a
decrease in the land carbon sink (Sitch et al., 2007; Wittig et al., 2007; Lombardozzi et al., 2015). Meanwhile,
vegetation helps reduce ambient ozone concentration through stomatal deposition (e.g., Kroeger et al., 2014).





However, the effect of such ozone-induced vegetation damage on ozone concentration itself, which thereby
completes the ozone-vegetation feedback loop, has not been examined before but is potentially significant in
modulating tropospheric ozone. This work uses a fully coupled land-atmosphere model to, for the first time,
quantify the impacts of ozone-vegetation coupling on surface ozone, and diagnoses the contributions from
various feedback pathways in terrestrial ecosystems.

Tropospheric ozone is mainly produced from the photochemical oxidation of carbon monoxide (CO),

methane ($CH_4$) and non-methane volatile organic compounds (VOCs) by hydroxyl radical (OH) in the presence
of nitrogen oxides ($NO_x \equiv NO + NO_2$). Vegetation plays various significant roles modulating surface ozone
concentration. Precursor gases of ozone have large anthropogenic and natural sources, including vegetation and
soil microbes for $CH_4$ and other VOCs. The most abundant single non-methane VOC species emitted by
vegetation is isoprene ($C_5H_8$), which acts as a major precursor for ozone formation in polluted, high-$NO_x$
regions, but eliminates ozone by direct ozonolysis or by sequestering $NO_x$ as isoprene nitrate in more pristine
environments (Fiore et al., 2011). The major sinks for tropospheric ozone include photolysis in the presence of
water vapor and uptake by vegetation (i.e., dry deposition, mainly through the leaf stomata). Vegetation,
therefore, plays a significant role in modulating ozone biogeochemically through dry deposition and biogenic
VOC emissions. Meanwhile, transpiration from vegetation can affect ozone by regulating the overlying
hydrometeorological environment. For instance, transpiration influences near-surface water vapor content,
which affects the chemical loss rate of ozone. Transpiration also controls surface temperature and mixing depth,
which can all influence the formation and dilution of ozone in the atmospheric boundary layer (Jacob and
Winner, 2009).

Vegetation not only affects but is also affected by surface ozone. Stomatal uptake of ozone by leaves

damages internal plant tissues, leading to severe damage to forest, grassland and agricultural productivity
(Ashmore, 2005; Karnosky et al., 2007; Ainsworth et al., 2012). Elevated ozone since the industrial revolution
is suggested to have reduced light-saturated photosynthetic rate and stomatal conductance by 11% and 13%,
respectively (Wittig et al., 2007). Modeling studies have also suggested that elevated ozone could decrease gross
primary production (GPP) by 4-8% in the eastern US and more severely so (11-17%) in several hot spots there
(Yue and Unger, 2014), and decrease transpiration rate globally by 2-2.4% (Lombardozzi et al., 2015), with
significant implications for climate. For instance, the ozone-induced reduction in the global land carbon sink is
shown to have an indirect radiative forcing of +0.62-1.09 W $m^{-2}$, which is comparable to the direct radiative
forcing of ozone as a greenhouse gas (0.89 W $m^{-2}$) and contributes to more pronounced warming (Sitch et al.,
2007). Changes in stomatal conductance also modify the land-atmosphere exchange of water and energy and
thus regional hydrometeorology (Bernacchi et al., 2011; Lombardozzi et al., 2015). In view of the important
roles vegetation plays in shaping tropospheric ozone, the above biogeochemical and biogeophysical effects
induced by ozone damage would affect not only weather and climate but also constitute important feedbacks
that ultimately affect ozone air quality itself.

In many land surface models, photosynthetic rate and stomatal conductance are highly coupled through

the computation within the Farquhar/Ball-Berry model (Farquhar et al., 1980; Ball et al., 1987; Bonan et al.,
2011). In global modeling studies on ozone-mediated vegetation changes and climate (Sitch et al., 2007; Collins
et al., 2010; Yue and Unger, 2014), the effects of ozone damage on photosynthesis and stomata are thus strongly
coupled to each other. Ozone uptake is assumed to directly affect photosynthetic rate, which in turn affects



stomatal conductance via changes in internal $CO_2$ concentration. However, recent studies have suggested that
separate modification of photosynthetic rate and stomatal conductance by cumulative ozone uptake in the
Community Land Model (CLM) leads to better representation of plant responses to ozone exposure
(Lombardozzi et al., 2012). This decoupling of ozone effects on photosynthesis and stomata is shown to
decrease water use efficiency of affected plants, but leads to an overall smaller impact of ozone on transpiration
and GPP than previously predicted.
Many climate-chemistry-biosphere modeling studies performed to date have demonstrated the
importance of the coevolution of climate, land cover and terrestrial ecosystems in air quality simulations and
predictions (Wu et al., 2012; Tai et al., 2013; Pacifico et al., 2015), but they have not taken into account the
potentially strong feedbacks arising from ozone damage on vegetation. For instance, ozone exposure can reduce
stomatal conductance and thus transpiration rate, which may modify the partition between latent and sensible
heat fluxes and lead to a cascade of meteorological changes, including lower humidity that reduces the chemical
loss rate of ozone, a thicker boundary layer that dilutes all pollutants, and higher temperature that enhances
ozone mainly through increased biogenic emissions and higher abundance of $NO_x$ (Jacob and Winner, 2009).
These transpiration-mediated pathways can be characterized as biogeophysical feedbacks as are commonly
known in the context of climate change, but here we prefer to call them simply "meteorological feedbacks" to
emphasize that they are effected through ozone-induced changes in the meteorological variables that ultimately
affect ozone. On the other hand, reduced dry deposition caused by lower stomatal conductance and a decline in
leaf area index (LAI) following ozone exposure can potentially increase ozone. The short-term impact of ozone
on foliage-level isoprene emission is still under debate (Fares et al., 2006; Calfapietra et al., 2007), but as
vegetation density (e.g., represented by LAI) declines due to chronic ozone exposure (Yue et al., 2014), isoprene
emission would likely decrease in the long term. These pathways directly involving plant biogeochemistry and
atmospheric chemistry can be collectively termed "biogeochemical feedbacks". Fig. 1 summarizes the
potentially important biogeochemical and meteorological feedbacks on surface ozone concentration, which are
expected to have ramifications for simulations and future projections of ozone air quality. Such feedbacks may
further alter atmospheric composition (e.g., aerosol and oxidant concentrations) and climate at large but remain
poorly characterized in an Earth system modeling framework.
In this study, we adopt and implement a semi-empirical scheme for ozone-induced vegetation damage
(Lombardozzi et al., 2015) into a coupled land-atmosphere model with fully interactive atmospheric chemistry
and biogeochemical cycles, and examine the resulting impacts on present-day simulations of tropospheric ozone
air quality with respect to observations. We perform sensitivity simulations to quantify the relative importance
of different biogeochemical and meteorological feedback pathways, elucidate the larger sources of uncertainties,
and make specific suggestions regarding Earth system model development.

**2 Methods**
**2.1 Model description**
This study investigates the impacts of ozone-vegetation coupling on ozone concentrations using the
Community Earth System Model (CESM), which includes several different model components representing the
atmosphere, land, ocean, and sea ice to be run independently or in various coupled configurations (Oleson et al.,
2010; Lamarque et al., 2012; Neale et al., 2013). We employ CESM version 1.2.2 with fully interactive





atmosphere and land components, but with prescribed ocean and sea ice consistent with the scenarios of
concern. For the atmosphere component, we use the Community Atmosphere Model version 4 (CAM4) (Neale
et al., 2013) fully coupled with an atmospheric chemistry scheme (i.e., CAM-Chem) that contains full
tropospheric $O_3$-$NO_x$-CO-VOC-aerosol chemistry based on the MOZART-4 chemical transport model (CTM)
(Emmons et al., 2010; Lamarque et al., 2012). The version of CAM-Chem simulates the concentrations of 56
atmospheric chemical species at a horizontal resolution of 1.9°×2.5° latitude-longitude and 26 vertical layers for
the atmosphere up to around 40 km.

For the land component, we use the Community Land Model version 4 (CLM4) (Oleson et al., 2010)

with active carbon-nitrogen biogeochemistry (CLM4CN), which contains prognostic treatment of terrestrial
carbon and nitrogen cycles (Lawrence et al., 2011). In CLM4, the Model of Emissions of Gases and Aerosols
from Nature (MEGAN) version 2.1 is used to compute biogenic emissions online as functions of changing LAI,
vegetation temperature, soil moisture and other environmental conditions (Guenther et al., 2012). For dry
deposition of gases and aerosols we use the resistance-in-series scheme in CLM4 as described in Lamarque et
al. (2012) with a further update of optimized coupling of stomatal resistance to LAI (Val Martin et al., 2014).
Evapotranspiration is calculated based on the Monin-Obukhov similarity theory and the diffusive flux-resistance
model with dependence on vegetation, ground and surface temperature, specific humidity, and an ensemble of
resistances that are functions of meteorological and land surface conditions (Oleson et al., 2010; Lawrence et al.,
2011; Bonan et al., 2011). Evapotranspiration is partitioned into transpiration, ground evaporation and canopy
evaporation, with updates from Lawrence et al. (2011), and is linked to photosynthesis via the computation of
stomatal resistance, as described below.

**2.2 Photosynthesis- stomatal conductance model and ozone damage parameterization**

The Farquhar/Ball-Berry model is used in CLM4CN to compute leaf-level photosynthetic rate and

stomatal conductance under different environmental conditions (Farquhar et al., 1980; Ball et al., 1987).  Leaf
photosynthetic rate, $A$ (μmol $CO_2$ m$^{-2}$ s$^{-1}$), is calculated as
$$A = \min (W_c, W_j, W_e) \tag{1}$$
where $W_c$ is the Ribulose-1,5-bisphosphate carboxylase (RuBisCO)-limited rate of carboxylation, $W_j$ is the light-
limited rate, and $W_e$ is the export-limited rate is $W_e$. Photosynthesis and stomatal conductance ($g_s$) are related by
$$g_s = \frac{1}{r_s} = m \frac{A}{c_s} \frac{e_s}{e_i} P_{\text{atm}} + b \tag{2}$$
where $g_s$ is the leaf stomatal conductance; $r_s$ is the leaf stomatal resistance (s m$^2$ μmol$^{-1}$); $m$ is the slope of the
conductance-photosynthesis relationship with values ranging from 5 to 9; $c_s$ is the $CO_2$ partial pressure at leaf
surface (Pa); $e_s$ is the vapor pressure at leaf surface (Pa); $e_i$ is the saturation vapor pressure inside the leaf (Pa);
$P_{\text{atm}}$ is the atmospheric pressure (Pa); and $b$ is the minimum stomatal conductance when $A = 0$, and is set to give
a maximum stomatal resistance of 20000 s m$^{-1}$ in CLM4 (Oleson et al., 2010).

Parameterization for the impact of ozone exposure on photosynthesis and stomatal conductance follows

the work of Lombardozzi et al., (2015), who tested the sensitivity of global ecosystem productivity and
hydrometeorology to ozone damage on vegetation using satellite phenology (i.e., prescribed LAI, canopy height,
etc.) and present-day ozone concentrations. The scheme uses two sets of ozone impact factors, one for
modifying photosynthetic rate and another for stomatal conductance independently. These factors account for
different plant groups, and are calculated based on the cumulative ozone uptake (CUO) under different levels of
chronic ozone exposure (Lombardozzi et al., 2013). CUO integrates ozone flux into leaves over the growing
season as
$\text{CUO} = \sum(k_{O_3}/r_s)\,[O_3]$ (3)
where $[O_3]$ is the surface ozone concentration computed from CAM-Chem in every time step, and $k_{O_3}$ is the
ratio of leaf resistance to ozone to leaf resistance to water. Ozone uptake is only cumulated during the growing
season when vegetation is the most vulnerable to air pollution episodes; growing season is defined as the period
in which total leaf area index (TLAI) > 0.5 (Lombardozzi et al., 2012). Ozone uptake only cumulates when the
ozone flux is above a critical threshold, 0.8 nmol $O_3$ m$^{-2}$ s$^{-1}$, to account for ozone detoxification by vegetation at
lower ozone levels (Lombardozzi et al., 2015). Three different plant groups are accounted for: evergreen,
deciduous, and crops/grasses. The ozone impact factors have empirical linear relationships with CUO such that
$F_{pO_3} = a_p \times \text{CUO} + b_p$ (4)
$F_{cO_3} = a_c \times \text{CUO} + b_c$ (5)
where $F_{pO_3}$ is the ozone damage factor multiplied to the photosynthesis rate ($A$), and $a_p$ and $b_p$ are slope and
intercept from empirical and experimental studies (listed in Table 1); $F_{cO_3}$ is the ozone damage factor multiplied
to stomatal conductance ($g_s$), and $a_c$ and $b_c$ are the corresponding slope and intercept (Table 1). The ozone
damage is applied to the optimal photosynthesis and stomatal conductance values, which are calculated
iteratively first without ozone damage, to allow the damage to be applied independently.

**2.3 Model experiments**

Incorporating the ozone-vegetation parameterization above into CLM4CN and coupling it with CAM-

Chem, we allow, for the first time, ecosystem structure (e.g., in terms of LAI and canopy height) to evolve in
response to ozone exposure but at the same time allow ozone concentration to evolve in response to such
ecosystem changes. Therefore, online ozone-vegetation coupling and feedback are included. We conduct four
sets of fully coupled land-atmosphere simulations: 1) a control case without ozone damage on vegetation
([CTR]); 2) simulation with both photosynthetic rate and stomatal conductance modified by ozone impact
factors (independently) ([PHT+COND]), following the approach of Lombardozzi et al (2015); 3) simulation
where we apply the ozone impact factor to photosynthetic rate only ([PHT]), but stomatal conductance is
calculated using the intact, optimal photosynthetic rate; and 4) simulation where we apply the ozone impact
factor to stomatal conductance only ([COND]), but photosynthetic rate is calculated using the intact stomatal
conductance. Simulations [PHT] and [COND], when compared with [PHT+COND], allow us to quantify the
relative contribution from each pathway. To determine the relative contribution of those pathways involving
biogenic emissions toward the overall ozone-vegetation feedback, we conduct an additional set of sensitivity
simulations with prescribed isoprene emission and MEGAN turned off: a control case with no MEGAN
(CTR_nM), and a simulation with modified photosynthesis and stomatal conductance but with no MEGAN
([PHT+COND_nM]). To determine the relative contribution of pathways involving dry deposition vs.
transpiration, we compare simulated results with that of Val Martin et al. (2014) who have used the similar





CAM-Chem-CLM framework but without ozone-vegetation coupling to test the sensitivity of ozone to
perturbations in dry deposition velocity.

All simulations are conducted for 20 years using year 2000 initial conditions and the corresponding

land cover data (e.g., land cover and land use types, satellite LAI, etc.). The first five years of outputs are treated
as spin-up and thus discarded in the analysis. We observe that the annual averages of key aboveground
ecosystem parameters such as LAI and ozone concentration come into a relatively steady state after 5 years. We
focus on changes in the 15-year northern summertime (JJA) averages for most of the variables in the rest of this
paper because this is the period when the growing season of the majority of global vegetation overlaps most
significantly with high-ozone season especially in the northern midlatitudes.

**3 Simulated ozone with and without ozone-vegetation coupling**

Figure 2(a) shows the 15-year mean summertime surface ozone concentration from the [PHT+COND]

simulation. The corresponding cumulative ozone uptake (CUO) used to affect vegetation is shown in Fig. 2(b).
Simulated ozone is generally higher in the northern midlatitudes than elsewhere, and is the highest over the
Mediterranean where solar radiation is particularly strong. CUO also has high values in Europe, but the overall
distribution does not exactly follow that of surface ozone concentration because CUO also depends on the
length of the growing season and stomatal conductance. CUO ranges between 20-70 mmol m$^{-2}$ over regions
with both high summertime ozone and high productivity. The simulated CUO is comparable in both magnitude
and spatial distribution with Lombardozzi et al., (2015), who used prescribed meteorology, ozone and
vegetation phenology with no active carbon-nitrogen cycle or atmospheric coupling, as opposed to this study.
This suggests that online ozone-vegetation coupling, which can modify ozone concentration substantially
depending on the region, lead to a similar pattern of ozone damage on vegetation to the case using prescribed
ozone. During the growing season, CUO is used to calculate the ozone impact factors that modify
photosynthetic rate and stomatal conductance according to Eq. (4) and (5) and parameter values listed in Table

1.

Figure 3 shows the differences in surface ozone concentration in different simulations from the control

case. Implementing ozone-vegetation coupling that includes simultaneous modification of photosynthetic rate
and stomatal conductance by ozone exposure (the [PHT+COND] case) increases mean surface ozone globally,
and significant increases by up to 4-6 ppbv are found over China, North America and Europe (Fig. 3a). Ozone
exposure is thus found to constitute a positive feedback loop via vegetation that ultimately enhances surface
ozone levels when ozone-vegetation coupling is accounted for.

The simulated increases in ozone levels due to ozone-vegetation coupling are significant when

compared with the possible impacts of 2000-2050 climate and land cover changes on surface ozone, which are
in the range of +1-10 ppbv (Jacob & Winner, 2009; Tai et al., 2013; Val Martin et al., 2015). These simulated
increases, however, slightly worsen the performance of CAM-Chem in reproducing ozone concentrations
against observations as seen in Fig. 4, which shows the model-observation comparison for the control case
(standard CAM-Chem-CLM with dry deposition improvement of Val Martin et al. (2014)) and the
[PHT+COND] case. The high-biases in CESM-simulated summertime surface ozone concentrations in North
America and Europe are a commonly acknowledged issue with CAM-Chem (Lamarque et al., 2012) and other
global and regional models (Lapina et al., 2014; Parrish et al., 2014). Inclusion of ozone-vegetation coupling in





the model further increases the normalized mean biases of the modeled results against three sets of observational
data: Clean Air Status and Trends Network (CASTNET) (1999-2001), Air Quality System (AQS) (1999-2001),
and European Monitoring and Evaluation Programme (EMEP) (1999-2001), from 18% to 22%, 31% to 35%,
14% to 22%, respectively. Given the sound theoretical and empirical basis of ozone damage on vegetation, this
further highlights the urgency to revise other model processes and modules relevant for ozone simulations.

**4 Attribution to different biogeochemical and meteorological feedback pathways**

Figures 3(b) and 3(c) show the differences in ozone for the cases where ozone damages stomatal

conductance alone and photosynthesis alone, respectively, noting that each of them is calculated using the
undamaged, intact values of the other variable. Comparison of Fig. 3(a) with (b)-(c) shows that the modification
of stomatal conductance by ozone uptake contributes more dominantly to the overall effect of ozone-vegetation
coupling (Fig. 3a). This suggests that, among the various feedback pathways that may influence surface ozone
(Fig. 1), those triggered by changes in stomatal conductance are generally more important than those associated
with photosynthesis or the associated changes in ecosystem production and structure including LAI, at least in
the modeling framework of this study. This is also supported by sensitivity simulations performed under the
same modeling framework but without ozone damage, in which a 50% of increase in LAI decreases
summertime surface ozone by on average 3 ppb, which is relatively small in comparison with the changes
following optimization of stomatal resistance (Val Martin et al., 2014). Indeed, the effect of modifying stomatal
conductance alone ([COND]; Fig. 3b) is slightly larger than the case of [PHT+COND] (Fig. 3a), where the
additional effect of modifying photosynthesis together with stomatal conductance would slightly offset the
overall positive feedback on ozone. It is noteworthy that this additional effect is, however, not consistent with
the effect of modifying photosynthesis alone ([PHT]; Fig. 3c), reflecting nonlinear interactions between
photosynthesis and stomatal conductance.

Figure 5 shows the differences in dry deposition velocity, transpiration rate and biogenic isoprene

emission between the [PHT+COND] and [CTR] simulations. Over China, Europe and North America, ozone
dry deposition velocity is lower (by up to ~20%) in [PHT+COND]. In these same regions but especially in the
eastern US, southern Europe and southern China, isoprene emission is significantly higher (by up to ~50%). In
addition, in similar regions but especially in central North America, the transpiration rate is reduced by ozone
exposure (by up to ~20%), which would reduce boundary-layer humidity, increase surface temperature, enhance
dry convection and thicken the boundary layer. In view of Fig. 1, all of these pathways may add to or offset each
other, leading to the overall ozone changes seen in Fig. 3(a). The sensitivity simulations and comparison with
Val Martin et al. (2014), which examined the sensitivity of simulated ozone to differences in dry deposition
schemes under essentially the same modeling framework, allow us to quantify more precisely which of these
pathways are more important as we discuss next.

Figure 6(a) shows the changes in surface ozone in the [PHT+COND_nM] minus CTR_nM simulations,

where we use prescribed biogenic emissions from the original control case (CTR) to drive ozone chemistry so
that we essentially shut down any feedback pathways involving biogenic emissions. A comparison between Fig.
6(a) and Fig. 3(a) shows that the changes in biogenic isoprene emissions account for ~0-60% of the ozone
increases over Europe, North America and China, while dry deposition and/or transpiration-driven
meteorological changes (excluding the temperature effect on isoprene emission) account for remaining ~40-





100%. We further show in Fig. 6(b) the theoretical changes in surface ozone by multiplying the dry deposition
changes in Fig 5(a) by the change in ozone concentration per unit change in dry deposition velocity from the
study of Val Martin et al. (2014). We find that the ozone changes in Fig. 6(a) and Fig. 6(b) are similar in
magnitude, suggesting that globally most of the non-isoprene-driven differences in ozone is driven by dry
deposition. Notable exceptions include the US Midwest and southeastern Europe, where higher mixing depth
following reduced transpiration might have partly offset the ozone positive feedback; and western Europe,
where the lower chemical loss rate following reduced transpired water might have further enhanced the positive
feedback.

The simulated general reduction in dry deposition velocity and transpiration rate (Fig. 5a and 5b) is

mostly due to increased stomatal resistance (Fig. 7a), i.e., reduced stomatal conductance, a direct response to
cumulative ozone uptake. The reduced dry deposition represents a positive biogeochemical feedback on ozone
(orange arrows in Fig. 1). The simulated increase in biogenic isoprene emission (Fig. 5c) is found to be mostly
driven by higher surface (thus vegetation) temperature (Fig. 7b) that results from lower transpiration rate and
latent heat flux (Fig. 7c). Therefore, this feedback loop involving biogenic emissions is indeed an indirect,
meteorological feedback that is also initiated by stomatal and transpiration changes (purple arrows in Fig. 1).

By including immediate ozone-vegetation coupling, we find a larger decline in transpiration rate (6.4%

globally) than in the offline, uncoupled land model results (2.0-2.4%) estimated by Lombardozzi et al. (2015).
On the other hand, although reduced photosynthesis and the resulting long-term changes in GPP and LAI (Fig.
7d-e) play a smaller role than reduced stomatal conductance in shaping simulated ozone (Fig. 3b-c), the impacts
are not negligible (up to 3 ppb), especially as these changes are also nonlinearly coupled to stomatal changes.
Photosynthetic rate decreases by up to 20% directly due to the ozone effect (Fig. 7f), which is quite similar both
in magnitude and spatial pattern to the results of Lombardozzi et al. (2015), but the corresponding GPP and LAI
changes are relatively small (~5% over regions concerned), likely reflecting the relaxation of nitrogen limitation
when photosynthesis is reduced. Grid-level GPP and LAI in certain areas increase despite reduced leaf-level
photosynthetic rate, likely reflecting more carbon allocation to leaves to compensate the reduced photosynthetic
rate, relaxation of nitrogen limitation, and enhanced vegetation temperature following reduced transpiration.

**5 Conclusions and discussion**

Tropospheric ozone is one the most hazardous air pollutants due to its harmful effects on human health

and damage to forest and agricultural productivity. Stomatal uptake of ozone by leaves reduces both
photosynthetic rate and stomatal conductance. These vegetation changes can induce a cascade of
biogeochemical and biogeophysical (or meteorological) effects (Fig. 1) that ultimately modulate climate, carbon
cycle and also feedback onto ozone air quality itself. The direct, biogeochemical feedback pathways include
reduced ozone dry deposition and biogenic VOC emissions. The indirect, meteorological feedback pathways are
facilitated by transpiration-driven changes in the meteorological environment that influence ozone formation
and removal. Many land surface modeling studies have estimated the direct effects of ozone on ecosystem
production and land-atmosphere water exchange (Yue and Unger, 2014; Lombardozzi et al., 2015), and
predicted a possible positive radiative forcing from the ozone-induced decline in the land-carbon sink (Sitch et
al., 2007). However, the potentially large feedback effects onto ozone concentration itself have not been
examined, which may have further ramifications for climate forcing because of the greenhouse effect of ozone.





In this study, we implement a semi-empirical parameterization of ozone damage on vegetation
(Lombardozzi et al., 2015) into the CESM (CAM4-Chem-CLM4CN) modeling framework to enable online
ozone-vegetation coupling so that vegetation variables can evolve in response to ozone exposure, and at the
same time simulated ozone concentration can respond to ecosystem changes. Our scheme modifies leaf-level
photosynthesis and stomatal conductance separately via the ozone impact factors, which are assumed to have
empirical linear relationships with cumulative ozone uptake and account for different plant groups. Sensitivity
simulations are conducted to determine the relative importance of different feedback pathways.
With ozone-vegetation coupling, surface ozone is simulated to be higher by up to 6 ppbv over Europe,
North America and China. This coupling effect is significant in view of the 2000-2050 effects of climate and
land cover changes on surface ozone (+1-10 ppbv) as found in previous work (Jacob and Winner, 2009; Tai et
al., 2013), and should be considered in future air quality projection studies. Reduced dry deposition velocity
following the modification contributes to ~40-100% and enhanced biogenic isoprene emission contributes to ~0-
60% of the higher ozone concentrations. The dry deposition-driven ozone increases (by up to 4 ppbv) arise
mostly from reduced stomatal conductance, and are consistent with the sensitivity of ozone to perturbations in
dry deposition velocity found by Val Martin et al. (2014). This pathway constitutes a significant positive
biogeochemical feedback on surface ozone. The other major feedback associated with enhanced isoprene
emission is mostly driven by higher vegetation temperature that results from lower transpiration rate. This
pathway constitutes an indirect, positive meteorological feedback on surface ozone. Depending on the region,
transpiration-driven meteorological changes such as lower humidity and deeper mixing depth may also
influence surface ozone. Transpiration rate is simulated to decrease by 6.4% globally, which is a larger change
compared with the decrease estimated by Lombardozzi et al. (2015) and suggests an augmented effect due to the
coupling between the atmosphere and ecosystems.
Modification of photosynthesis and further long-term changes in ecosystem productivity and structure,
including LAI changes, are found to play a smaller role in contributing to the ozone-vegetation feedbacks than
direct stomatal changes, but are not insignificant (up to +3 ppbv). The simulated changes in LAI (less than 5%)
in this study are similar in magnitude to that by Yue et al. (2015), who included an active carbon cycle though
using Yale Interactive terrestrial Biosphere (YIBs) model with a different ozone-vegetation parameterization.
However, prognostic treatment of the carbon cycle and LAI calculation in CLM4CN are still known to be
problematic, with large uncertainties and biases in the estimation of global carbon fluxes (Sun et al., 2012),
arising from incomplete model parameterization and from uncertainty in photosynthetic parameters (Bonan et
al., 2011). It is not surprising that changes in GPP as simulated here do not replicate the results of Lombardozzi
et al. (2015), in which vegetation phenology is prescribed and the carbon and nitrogen cycles are not active
(CLM4.5SP). Implementing ozone damage on vegetation in a model with more sophisticated and realistic
representation of prognostic carbon-nitrogen cycle is highly warranted, so that the possible effects of ozone-
induced long-term ecosystem changes can be examined more fully.
Large variability in the responses of different plants to ozone leads to considerable uncertainties in any
global-scale studies (Lombardozzi et al., 2013). The model results could be improved with more detailed plant-
type-specific ozone damage parameterization, including better estimates of plant vulnerability to ozone that will
help refine the ozone uptake thresholds (Lombardozzi et al., 2015). An important caveat of this study is the
consideration of only three plant groups to generalize the responses of global vegetation to ozone exposure



because data are largely unavailable for other plant groups. Another potential caveat is the uncertainty and lack
of cross-validation in hydrometeorological simulations with respect to the ozone phytotoxicity scheme we
newly implement, as we only focus on vegetation and atmospheric chemical changes in this study. Although
most simulated vegetation variables are consistent with previous work, the changes in simulated vegetation
temperature from ozone-vegetation coupling are not small (by up to +2°C) (Fig. 7b) and they result in quite
substantial changes in isoprene emission, suggesting the need for further tuning of hydrometerological processes
in the model. In general, we have the highest confidence in the quantification of the biogeochemical pathway
via stomata-driven deposition changes, which is straightforward and accounts for the majority of the ozone-
vegetation feedbacks. On the other hand, the meteorological feedbacks introduce strong nonlinearity in the
interactions between atmospheric chemistry and vegetation that is more difficult to isolate and understand.
Parameterizing the ozone-vegetation coupling in a standalone chemical transport model with prescribed
meteorology could be particularly helpful to more confidently separate between the effects of biogeochemical
vs. meteorological feedbacks. This knowledge will be important in projecting the impacts of future climate and
land cover changes on ozone air quality and climate feedbacks in the coming decades.

**Acknowledgment**
This work was supported by the Early Career Scheme (Project #: 24300614) of the Research Grants
Council of Hong Kong given to the principal investigator, Amos P. K. Tai. We also thank the Information
Technology Services Centre (ITSC) at The Chinese University of Hong Kong for their devotion in providing the
necessary computational services for this work.

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





Table 1. Slopes (per mmol m$^{-2}$) and intercepts (unitless) used to calculate ozone impact
factors in Eq. (4) and (5), following Lombardozzi et al. (2015).

|  | Photosynthesis | | Conductance | |
|---|---|---|---|---|
| Plant group | Slope ($a_p$) | Intercept ($b_p$) | Slope ($a_c$) | Intercept ($b_c$) |
| Broadleaf | 0 | 0.8752 | 0 | 0.9125 |
| Needleleaf | 0 | 0.839 | 0.0048 | 0.7823 |
| Crops and grasses | -0.0009 | 0.8021 | 0 | 0.7511 |







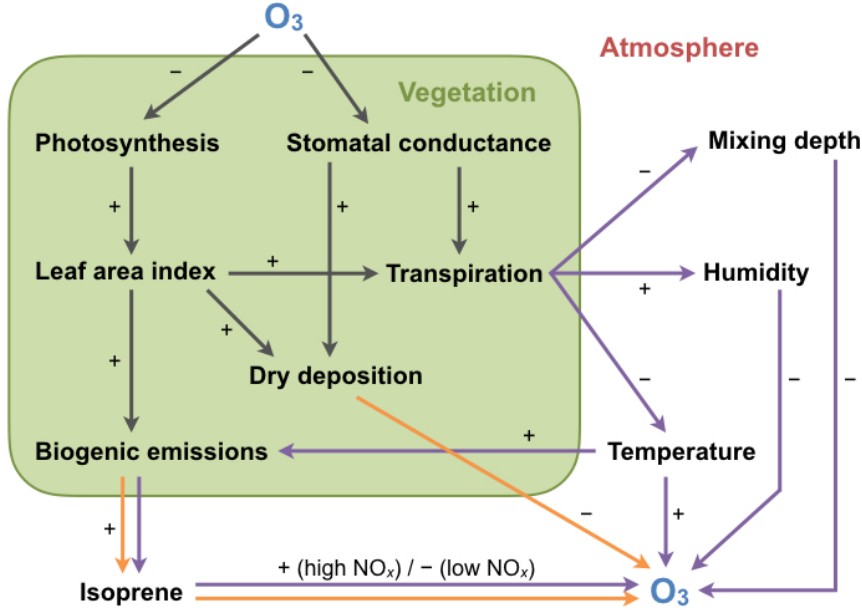

Figure 1. Possible pathways of ozone-vegetation coupling and feedbacks. The sign on each arrow indicates the sign of

correlation or effect of one variable with or on another variable; the product of all signs along a given pathway indicates the

overall sign of feedback. Orange arrows indicate biogeochemical feedbacks (i.e., via modulating atmospheric chemistry

directly); purple arrows indicate meteorological feedbacks (i.e., via modifying the meteorological environment). We focus

only on processes that directly affect ozone; meteorological feedbacks on photosynthesis and stomatal conductance are

included in the model but not emphasized in this figure.





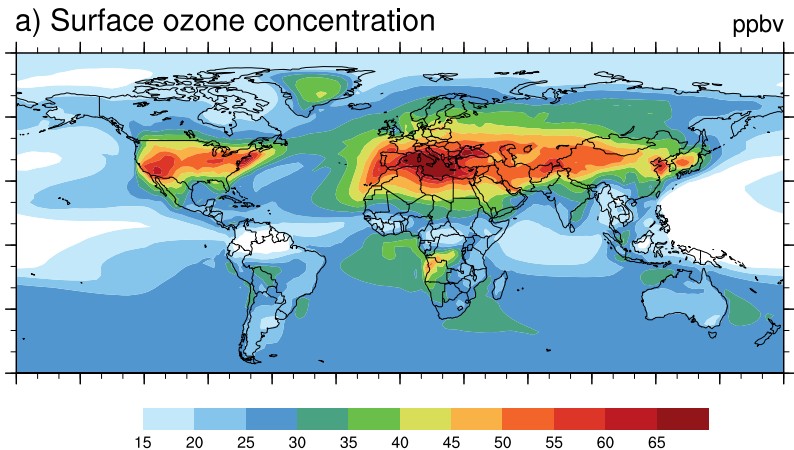

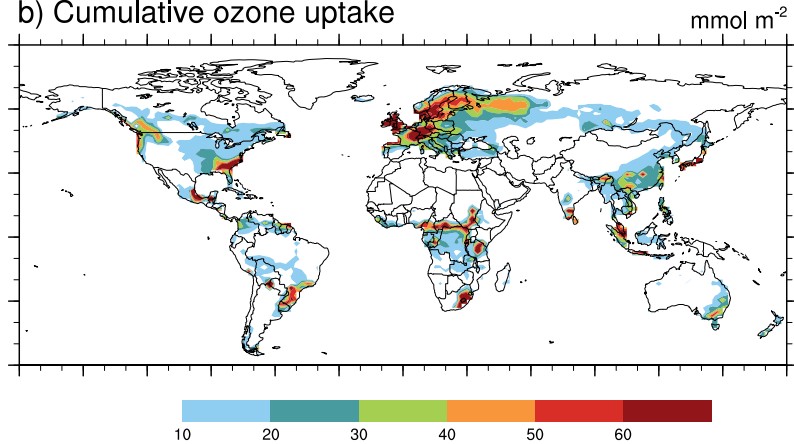


Figure 2. (a) Mean summertime (JJA) surface ozone concentration and (b) cumulative ozone uptake (CUO) from the

[PHT+COND] case, where ozone uptake simultaneously modifies both photosynthetic rate and stomatal conductance.

Results are averaged over the last 15 years of simulations.





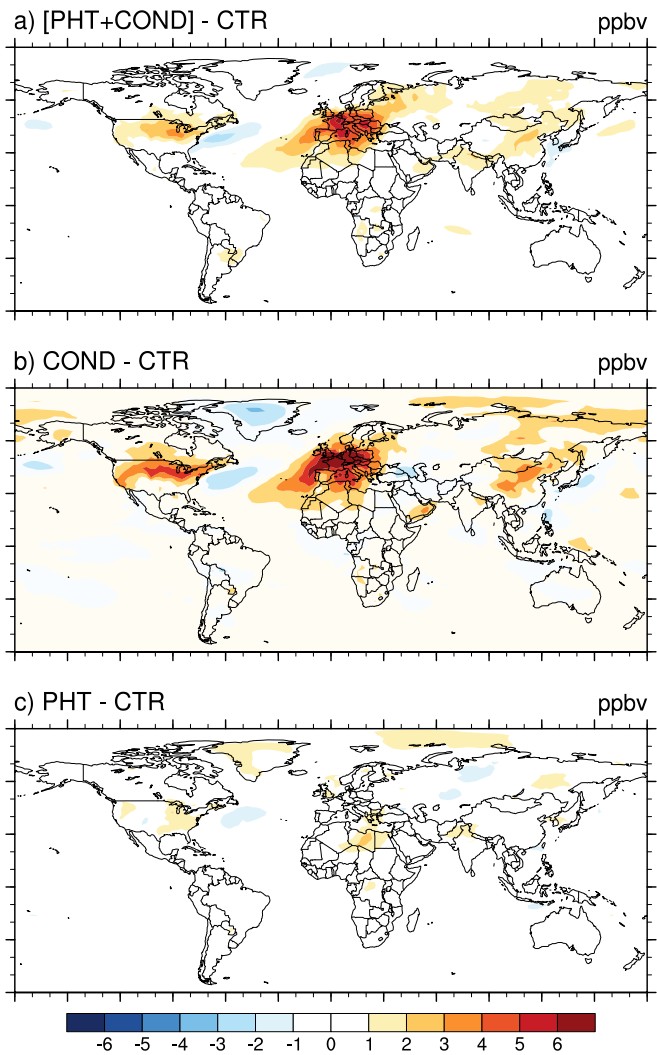


Figure 3. Changes in summertime surface ozone concentrations in different simulations: (a) the case where both
photosynthetic rate and stomatal conductance are modified by ozone uptake; (b) modified photosynthetic rate only; and (c)
modified stomatal conductance only, all relative to the control case (CTR).





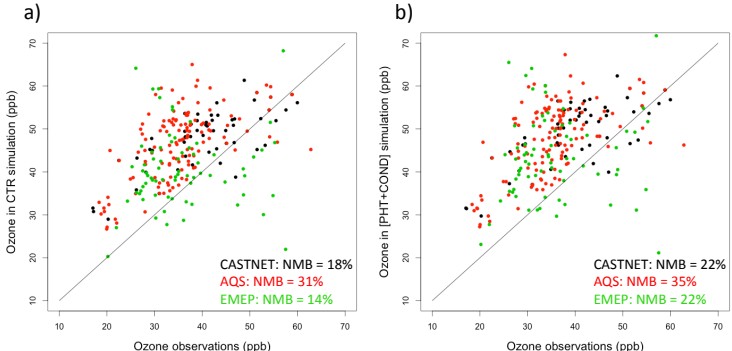


Figure 4. Scatterplots of simulated summertime ozone concentration in (a) the control case (CTR); and (b) the case where
both photosynthesis and conductance are modified by ozone uptake ([PHT+COND]), versus observed average values from
the Clean Air Status and Trends Network (CASTNET) (1999-2001), Air Quality System (AQS) (1999-2001), and European
Monitoring and Evaluation Programme (EMEP) (1999-2001). Normalized mean biases (NMB) are also shown.



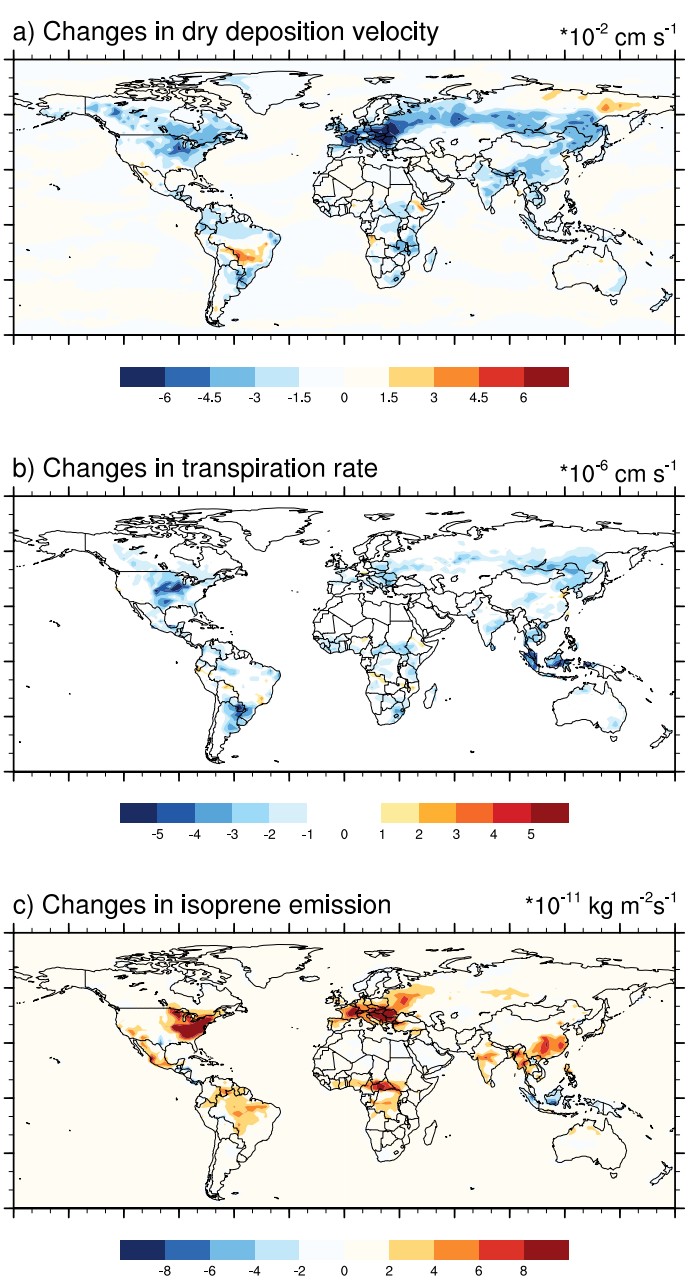

Figure 5. Changes in (a) dry deposition velocity, (b) transpiration rate and (c) isoprene emission in the [PHT+COND] case,
where both photosynthetic rate and stomatal conductance are modified by ozone uptake, relative to the control case (CTR).





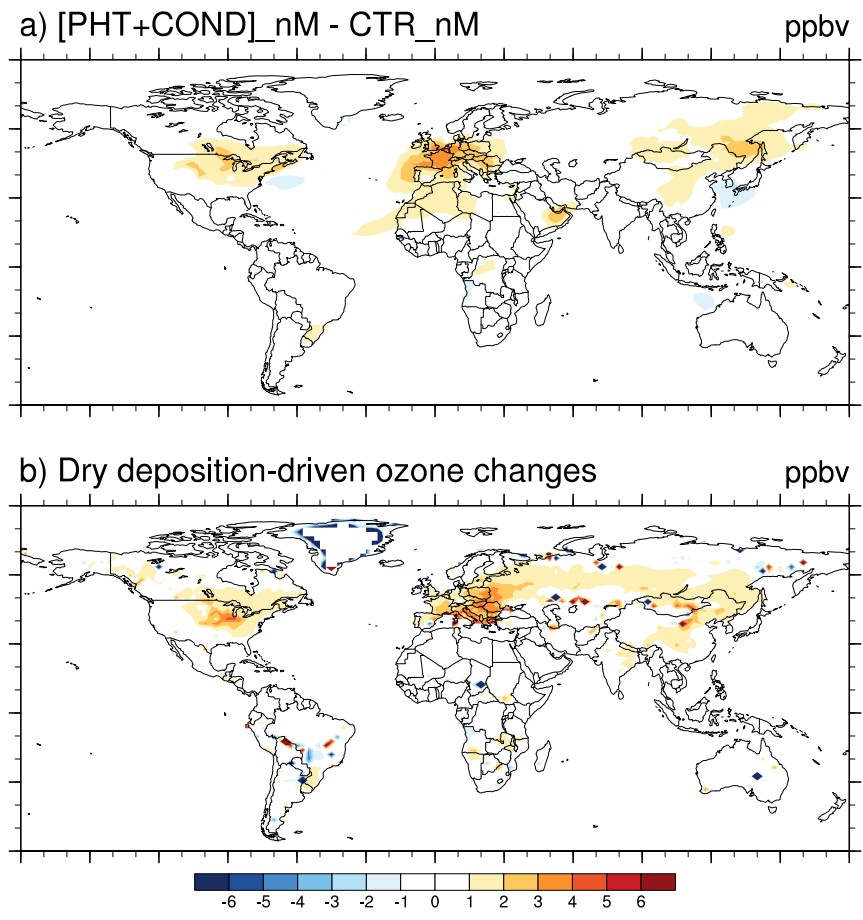


Figure 6. Changes in surface ozone concentration in: (a) the case where both photosynthesis and stomatal conductance are


modified by ozone uptake, but with prescribed isoprene emission from the original control case (CTR) by turning off


MEGAN; and (b) theoretical changes calculated by multiplying our simulated dry deposition changes with the change in


ozone concentration per unit change in dry deposition from Val Martin et al. (2014), which did not include ozone damage on


vegetation.






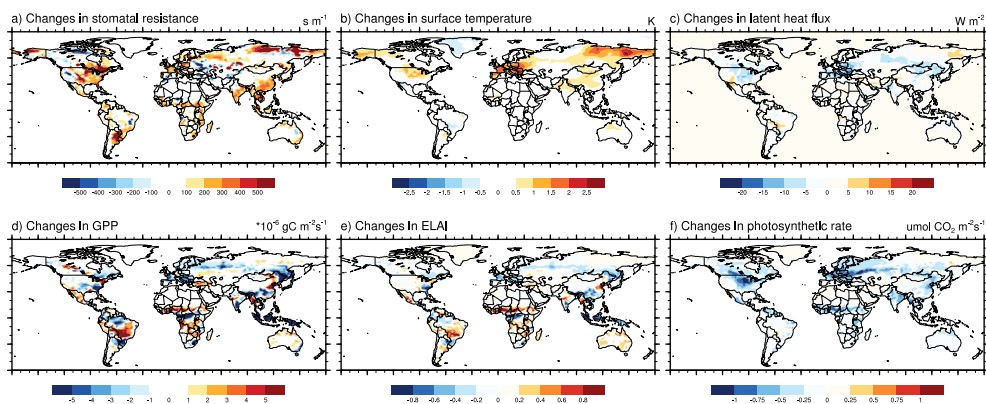


Figure 7. Changes in (a) stomatal resistance, (b) surface temperature, (c) latent heat flux, (d) gross primary production

(GPP), (e) effective leaf area index (ELAI) and (f) photosynthetic rate in the [PHT+COND] case, where both photosynthetic

rate and stomatal conductance are modified by ozone uptake, relative to the control case (CTR).