# Peer review of "Effects of ozone-vegetation coupling on surface ozone air 1"

_Atmospheric Chemistry and Physics, 2016_

## Short Comment (SC1) · 7 Sep 2016

I would like to give credits to the work done in this paper. A relatively complete modeling framework for global ozone levels is presented, including important feedbacks related to ozone damage. This work would be a nice addition to existing literature. However, I would like to share two small comments.

On lines 88-92 the authors summarize several important feedbacks of ozone damage on the ozone concentration itself. One of the feedbacks is that a decrease in stomatal conductance increases the boundary layer height, therefore reducing the in-situ ozone concentration. I would like to add that another consequence is the increase of entrainment, which can have either a positive or negative impact on surface ozone

concentrations.

Moreover, in the 'Model experiments' section the authors describe their different simulations. If I understand correctly, the PHT+COND run includes ozone impact factors for both the stomatal conductance and photosynthetic rate. Moreover, the photosynthetic rate affected by ozone uptake is used to calculate the stomatal conductance. So in principle there is a double impact on the stomatal conductance. In addition, the authors have chosen to do a simulation which includes an ozone impact factor for the photosynthetic rate, while calculating the stomatal conductance without such impact factor and with an intact photosynthetic rate. And vice versa for stomatal conductance (lines 184-188). I wonder why they use an intact photosynthetic rate as this simulation then has no physical meaning. I would rather suggest to use the photosynthetic rate corrected for ozone uptake, but setting the ozone impact factor for the stomatal conductance to 'no impact'. This would give a more realistic idea of the impact of the two separate pathways.
* * *

---

## Referee Comment (RC1) · Anonymous Referee #2 · 26 Sep 2016

This solid technical paper assesses the impact of ozone-vegetation coupling on surface ozone concentrations using a state of the art global Earth system model (CESM). The study builds on the recent informative work of Val Martin et al., GRL, 2014. A suite of sensitivity simulations is performed to unravel the relative roles of the modified biometeorological drivers in causing the altered surface ozone concentrations. The main overall conclusion is that surface ozone concentrations in the mid-latitude polluted temperate zone regions of Europe and North America are up to 6ppbv higher when the ozone-vegetation coupling is incorporated in the model framework. The sensitivity simulations indicate that the elevated surface ozone concentrations are mostly due to reduced dry deposition (stomatal closure on ozone damage) and increased iso-

prene emission (higher leaf temperatures due to reduced transpiration). Considerable effort and hard work has gone into conducting this challenging set of CESM simulations. The paper represents a valuable contribution to the literature in this emerging multidisciplinary field and deserves to be published in ACP once the following issues have been addressed. The findings may have important consequences in large-scale air quality modeling.

1. The main concern is the ozone damage function itself. The community accepts that flux-based damage schemes are more realistic than concentration schemes. However, it appears from the empirical parameters in Table 1 that in most cases (except photosynthesis for crops/grasses and stom. cond. for needleleaf) that the damage is independent of the CUO. Once the CUO has exceeded the threshold, the level of damage to photosynthesis and stom. cond. remains constant. This damage function does not seem realistic? Surely the level of plant physiological damage does depend (strongly!) on CUO. In their model, ozone vegetation damage for broadleaf biome is independent of CUO? More justification and explanation needs to be given as to the lack of dependence on CUO.

2. The results indicate that changes to photosynthesis have almost negligible impacts on the surface ozone concentrations. Isoprene emission is tightly coupled to photosynthesis (70-90%) that provides the energy and precursors for isoprene production. The MEGAN model does not include any direct connection to photosynthesis rate. Therefore, any influence of altered photosynthetic rate cannot change isoprene emission in this model framework. For example, ozone-induced photosynthetic reductions likely reduce isoprene emission and could potentially offset the temperature-related increases. Should a photosynthesis-based isoprene emission model have been used in this study to allow for all impacts on surface ozone? The current model misses this potentially important feedback. Are the surface ozone increases merely an artifact of the MEGAN model (or up to the 60% contribution)? Have the 6ppbv increases been largely overestimated?

3. A reference that may be helpful: Tiwari et al., Ozone damage, detoxification and the role of isoprenoids - new impetus for integrated models, http://dx.doi.org/10.1071/FP15302

4. Uncertainty ranges need to be provided e.g. on the 6 ppbv enhancements in surface ozone. These ranges could be based on 15-yr interannual variability from the multi-year simulations.

5. Figure 3 does not show substantial surface ozone feedbacks in China even though the Abstract (and text) claims up to 6 ppbv there.

6. It is interesting that inclusion of the ozone-vegetation coupling worsens the evaluation of simulated ozone against AQ measurement networks: "this further highlights the urgency to revise other model processes and modules relevant for ozone simulations." What are the other model processes that need to be revised? How large are the uncertainties in the surface ozone feedback estimates provided here, given that the damage function is based on only 3 biome types, is independent of CUO and the isoprene emission scheme is independent of photosynthetic rate?

7. Page 8, Line 298: "likely reflecting the relaxation of nitrogen limitation when photosynthesis is reduced". Needs more explanation in the text. What is "relaxation of nitrogen limitation"?

8. Page 9, Line 334: "Transpiration rate is simulated to decrease by 6.4% globally, which is a larger change compared with the decrease estimated by Lombardozzi et al. (2015) and suggests an augmented effect due to the coupling between the atmosphere and ecosystems." and discussed earlier in the paper. Why is the transpiration response 3 times larger in this work than in Lombardozzi et al., 2015? What exactly about the process of using coupled versus fixed ozone leads to this much larger impact?

9. Figure 2. I'm struggling to understand the CUO plot. Why is CUO relatively low in the mid-west U.S. crop belt, and e.g. very high in the UK where ozone concentrations

are rather low. In some ways, CUO plot is inverse of surface concentration plot, which is understandable on the basis of the deposition sink. But, why are you even showing CUO when the ozone vegetation damage functions are independent of CUO (with minor exceptions discussed in point (1) above)?

10. Page 8, Line 314: "which may have further ramifications for climate forcing because of the greenhouse effect of ozone." Increases in surface ozone have zero longwave radiative forcing because of the lack of thermal contrast with the surface. Ozone longwave radiative forcing is about changing ozone in the upper tropical troposphere.

11. Figure 5(c). The only isoprene decreases are in tropical rainforest SE Asia regions. What causes these reductions localized to this region? Other tropical rainforest areas in S. America/Amazon and central Africa show increases.

12. Figure 6(b). The dry-deposition driven ozone changes plot shows random sporadic grid cells with very high positive and negative values. What is causing these very high responses in a few random grid boxes? Would it help to show responses that are only statistically significant with 95% C.I. only grid cells, or similar? Otherwise, the results are technically unconvincing.

Minor comments 1. Abstract. Please remove "per se" and include quantitative description of statistical significance. 2. Page 5, Line 162. What is the time-step in the coupled land-atmosphere model? 3. Page 8, Line 310: "Many land surface modeling studies have estimated the direct effects of ozone on ecosystem production and land-atmosphere water exchange (Yue and Unger, 2014; Lombardozzi et al., 2015), and predicted a possible positive radiative forcing from the ozone-induced decline in the land-carbon sink (Sitch et al., 2007).". I suggest to change "Many" with "A few". 2-3 studies is not many. 4. Only use "significant" if you actually provide a quantitative statistical significance. 5. Need to include "+/-" in Figure 1 schematic e.g. for photosynthesis -> LAI as the model shows some increases in LAI when photosynthesis is reduced.

---

## Referee Comment (RC2) · Anonymous Referee #1 · 27 Sep 2016

INTRODUCTION:

This paper discusses the impacts of ozone on ecosystems in the framework of a modelling study. An ozone plant damage parametrization is implemented in CESM and fully coupled to the land surface and atmospheric composition components. The consequences of ozone ecosystem damages is investigated with regards to ozone surface concentrations, i.e., from an air quality perspective. A systematic analysis of ozone surface concentrations, dry deposition velocities, stomatal resistances, surface temperatures, latent heat fluxes, etc. is presented.

GENERAL REMARKS

The effect of ozone on ecosystems and its consequences for ecosystem productivity, dry deposition fluxes and ozone surface concentrations has been a hot topic for some time. This is mainly due to the seizable impacts that are expected on air quality and climate. Ozone can have a substantial control over the removal and/or emission of key atmospheric compounds such as biogenic VOC, ozone itself or water vapour. In the latter case, the impact on evapotranspiration is critical to the climate on a local to regional scale and the short term. This is compounded by the control of ozone over the carbon cycle through damages to photosynthetic productivity which has consequences for climate on the large scale and the long term. For these reasons I consider the paper a useful and interesting contribution to the current discussion. It may not be the absolute first study to apply fully coupled ozone parametrizations (c.f., e.g., Pacifico et al., 2015) but it is certainly amongst the pioneering studies and looks at it from a particular angle.

Largely, the paper is well written, accessible and nicely structured. In general, the figures are quite helpful and complement the text. There are a few changes I would recommend, though: 1) In case of figures 3, 5, 6 and 7 I would prefer to see relative changes rather than absolute changes. Maybe these figures could just be amended by plots of relative changes. 2) figure 4 does not seem to be discussed anywhere in the text. 3.) In case of figure 6 b I am not quite sure I understand the diagnostic. It seems to me this is some kind of semi-offline approach. Why is the "change in ozone concentration per unit change in dry deposition" required from another study? Is it not possible to diagnose the ozone change directly from the model simulations? At least, this should be explained more clearly.

For each experiment about 15 years of data are available, yet the authors only discuss means. There is a good opportunity to study inter-annual variability and add some statistics with regard to statistical robustness and significance of the findings. This is a somewhat serious omission in the current version of the paper which is the main and almost only reason why I recommend major revisions. However, this should be quite

easy to do.

SPECIFIC COMMENTS

p2l66-69: the number for the direct ozone RF of 0.89 W m-2 to me seems excessively high. If I recall well, the IPCC Assessment Report 5 quotes an ozone RF +0.35 (0.15–0.55) W m-2. Can you please clarify this number.

p4l146: "... , and W_e is the export-limited rate is W_e."; please revise

p4l158: the "cumulative ozone uptake" is abbreviated with CUO, however, I would find COU the more immediate choice. Or, if CUO is prefered then describe it as "cumulative uptake of ozone".

p5l162: "computed from CAM-Chem at every time step"; please revise

p5l164: "... when vegetation is most vulnerable to ..."

p7l238-239: the authors argue that "Given the sound theoretical and empirical basis of ozone damage on vegetation, this further highlights the urgency to revise other model processes and modules relevant for ozone simulations." However, I think this high confidence in our understanding of the ozone-ecosystem effects is not yet justified and I would not be prepared to modify other process models based on these findings. I think much more research on process understanding is needed and I would like to see this statement removed.

p8l275-277: As mentioned above in connection with figure 6, I don't quite understand the line of argument here. Why can the ozone impact on its own deposition not be diagnosed directly from the model simulations? Please explain further.

RECCOMENDATION

I recommend to reconsider the paper after major revisions. As I already stated above, this is only to include some analysis of the results with regard to statistical significance and robustness and/or some uncertainty analysis.

---

## Author Comment (AC1) · 13 Dec 2016

Responses to Reviewers on "Effects of ozone-vegetation coupling on surface ozone air quality via biogeochemical and meteorological feedbacks" by M. Sadiq, A. P. K. Tai, D. Lombardozzi and M. Val Martin. (MS No.: acp-2016-642)

Our point-by-point responses are provided below. The reviewers' comments are *italicized*, our new/modified text is highlighted in **bold**, and highlighted in blue in the manuscript.

Response to Referee #1

*INTRODUCTION:*

*This paper discusses the impacts of ozone on ecosystems in the framework of a modelling study. An ozone plant damage parametrization is implemented in CESM and fully coupled to the land surface and atmospheric composition components. The consequences of ozone ecosystem damages is investigated with regards to ozone surface concentrations, i.e., from an air quality perspective. A systematic analysis of ozone surface concentrations, dry deposition velocities, stomatal resistances, surface temperatures, latent heat fluxes, etc. is presented.*

*GENERAL REMARKS*

*The effect of ozone on ecosystems and its consequences for ecosystem productivity, dry deposition fluxes and ozone surface concentrations has been a hot topic for some time. This is mainly due to the seizable impacts that are expected on air quality and climate. Ozone can have a substantial control over the removal and/or emission of key atmospheric compounds such as biogenic VOC, ozone itself or water vapour. In the latter case, the impact on evapotranspiration is critical to the climate on a local to regional scale and the short term. This is compounded by the control of ozone over the carbon cycle through damages to photosynthetic productivity which has consequences for climate on the large scale and the long term. For these reasons I consider the paper a useful and interesting contribution to the current discussion. It may not be the absolute first study to apply fully coupled ozone parametrizations (c.f., e.g., Pacifico et al., 2015) but it is certainly amongst the pioneering studies and looks at it from a particular angle.*

*Largely, the paper is well written, accessible and nicely structured. In general, the figures are quite helpful and complement the text. There are a few changes I would recommend, though: 1) In case of figures 3, 5, 6 and 7 I would prefer to see relative changes rather than absolute changes. Maybe these figures could just be amended by plots of relative changes.*

We agree that showing relative changes is useful to evaluate the relative impacts of the implemented scheme. We now include figures for the relative changes in the same set of variables in Fig. 5 and 7 in the appendix (Fig. S3 and S4) for the readers' reference. We decide not to include them in the main text because the spatial distribution maps would be dominated by the apparently very large relative changes that indeed happen in "unimportant" places with low values of the corresponding variables to begin with. In contrast, the absolute changes may be more useful in highlighting regions where the impacts are practically important.

[Figure]

**Figure S3. Percentage changes in (a) dry deposition velocity, (b) transpiration rate and (c) isoprene emission in the [PHT+COND] case, where both photosynthetic rate and stomatal conductance are modified by ozone uptake, relative to the control case (CTR).**

[Figure]

**Figure S4. Percentage changes in (a) stomatal resistance, (b) surface temperature, (c) latent heat flux, (d) gross primary production (GPP), (e) effective leaf area index (ELAI) and (f) photosynthetic rate in the [PHT+COND] case, where both photosynthetic rate and stomatal conductance are modified by ozone uptake, relative to the control case (CTR).**

Moreover, for Fig. 3 and 6, in which we show differences in surface ozone concentrations, showing absolute values in the main text is considered more helpful for readers to understand the significance of the impacts, especially in comparison with the potential impacts of climate change or land use change alone on ozone, which are often reported in absolute terms [e.g., Jacob and Winner, 2009]. We also show the absolute values of summertime surface ozone concentration in Fig. 2 for easy comparison. For the readers' reference, we also add the relative changes of Fig. 3 as Fig. S2:

"Figure 3 shows the differences in surface ozone concentration in different simulations from the control case **(corresponding relative changes shown in supplemental Fig. S2).**" (p6 l223-224)

[Figure]

**Figure S2. Percentage changes in summertime surface ozone concentrations in different simulations: (a) the case where both photosynthetic rate and stomatal conductance are modified by ozone uptake; (b) modified photosynthetic rate only; and (c) modified stomatal conductance only, all relative to the control case (CTR).**

We now also add in the main text:

"Figure 5 shows the differences in dry deposition velocity, transpiration rate and biogenic isoprene emission between the [PHT+COND] and [CTR] simulations **(relative changes shown in supplemental Fig. S3)**. Over China, …" (p7 l266-268)

"Therefore, this feedback loop involving biogenic emissions is indeed an indirect, meteorological feedback that is also initiated by stomatal and transpiration changes (purple arrows in Fig. 1). **Relative changes in variables shown in Fig. 7 are included in supplemental Fig. S4.**" (p8 l298-300)

*2) figure 4 does not seem to be discussed anywhere in the text.*

Figure 4 is discussed in Sect. 3 (p6 l232-236).

*3) In case of figure 6 b I am not quite sure I understand the diagnostic. It seems to me this is some kind of semi-offline approach. Why is the "change in ozone concentration per unit change in dry deposition" required from another study? Is it not possible to diagnose the ozone change directly from the model simulations? At least, this should be explained more clearly.*

It is computationally challenging to decouple dry deposition, which is always computed online, from other land processes in our current model configuration. Dry deposition velocities cannot be readily read into CESM as offline inputs. Perturbing stomatal conductance to affect dry deposition also automatically affects meteorological conditions via transpiration, complicating the effects on ozone. We therefore use a separate set of experiments done by Val Martin et al. [2014] that compared between simulated ozone concentrations under two different dry deposition schemes to estimate the inherent sensitivity of surface ozone to dry deposition velocity in CESM, which is found to be mostly linear in the range of changes considered. We then use such sensitivity to calculate the theoretical changes in ozone caused by reduced deposition in [PHT+COND] case, which should give us a better estimate for the range of magnitude of the deposition-mediated effect alone without complication from other online factors. We now explain this in the text in Sect. 4:

"We further show in Fig. 6(b) the theoretical changes in surface ozone by multiplying the dry deposition changes in Fig. 5(a) by the change in ozone concentration per unit change in dry deposition velocity from the study of Val Martin et al. (2014**), which provided the sensitivity of simulated ozone to perturbed dry deposition only without the complication of stomatal coupling with hydrometeorology**. We find that…" (p8 l284-287)

*For each experiment about 15 years of data are available, yet the authors only discuss means. There is a good opportunity to study inter-annual variability and add some statistics with regard to statistical robustness and significance of the findings. This is a somewhat serious omission in the current version of the paper which is the main and almost only reason why I recommend major revisions. However, this should be quite easy to do.*

We have tested the statistical significance (simple *t*-test, *p*-value = 0.10) of the differences in surface ozone concentrations using the 15-year time series of summertime mean data, and now mark the grid cells with statistically significant changes in Fig. 3 and Fig. 6(a).

[Figure]

Figure 3. Changes in summertime surface ozone concentrations in different simulations: (a) the case where both photosynthetic rate and stomatal conductance are modified by ozone uptake; (b) modified photosynthetic rate only; and (c) modified stomatal conductance only, all relative to the control case (CTR). **Stippling with dots indicates significant changes at 90% confidence from Student's *t* test.**

[Figure]

Figure 6. Changes in surface ozone concentration in: (a) the case where both photosynthesis and stomatal conductance are modified by ozone uptake, but with prescribed isoprene emission from the original control case (CTR) by turning off MEGAN **(stippling with dots indicates significant changes at 90% confidence from Student's *t* test)**; and (b) theoretical changes calculated by multiplying our simulated

dry deposition changes with the change in ozone concentration per unit change in dry deposition from Val Martin et al. (2014), which did not include ozone damage on vegetation.

*SPECIFIC COMMENTS*

*p2 l66-69: the number for the direct ozone RF of 0.89 W m$^{-2}$ to me seems excessively high. If I recall well, the IPCC Assessment Report 5 quotes an ozone RF +0.35 (0.15– 0.55) W m$^{-2}$. Can you please clarify this number.*

These mentioned ozone-induced radiative forcings are by the end of 21$^{st}$ century, rather than for the present day. We added '**by 2100**' to this statement (p2 l66-67). 0.89 W m$^{-2}$ is the mean direct radiative forcing by 2100 from 11 atmospheric chemistry models (Sitch et al., 2007).

*p4 l146: "... , and W_e is the export-limited rate is W_e."; please revise*

(p4 l148) Revised as suggested.

*p4 l158: the "cumulative ozone uptake" is abbreviated with CUO, however, I would find COU the more immediate choice. Or, if CUO is prefered then describe it as "cumulative uptake of ozone".*

CUO is preferred and consistent with a previous paper that formed the foundation of methodology used in this study (Lombardozzi et al., 2013), so we revised the description to be "**cumulative uptake of ozone**" (p5 l160, p6 l210, p8 l295, p8 l335).

*p5 l162: "computed from CAM-Chem at every time step"; please revise*

(p5 l164) Revised as suggested.

*p5 l164: "... when vegetation is most vulnerable to ..."*

(p5 l166) Revised as suggested.

*p7 l238-239: the authors argue that "Given the sound theoretical and empirical basis of ozone damage on vegetation, this further highlights the urgency to revise other model processes and modules relevant for ozone simulations." However, I think this high confidence in our understanding of the ozone-ecosystem effects is not yet justified and I would not be prepared to modify other process models based on these findings. I think much more research on process understanding is needed and I would like to see this statement removed.*

We agree with the reviewer. There is still large uncertainty in the ozone-vegetation parameterization scheme. We removed the statement as suggested, and added following argument:

"**Although there remains considerable uncertainty in the parameterization of ozone-vegetation coupling and in ozone simulations by Earth system models, we show that including ozone damage in a coupled climate-chemistry-biosphere framework can have a potentially significant impact on surface ozone simulations.**" (p7 l244-247).

*p8 l275-277: As mentioned above in connection with figure 6, I don't quite understand the line of argument here. Why can the ozone impact on its own deposition not be diagnosed directly from the model simulations? Please explain further.*

See our response to point (3) above.

*RECCOMENDATION*

*I recommend to reconsider the paper after major revisions. As I already stated above, this is only to include some analysis of the results with regard to statistical significance and robustness and/or some uncertainty analysis.*

The manuscript has been revised, as detailed above, in response to the thoughtful comments and requests from the reviewer.

Responses to Referee #2

*This solid technical paper assesses the impact of ozone-vegetation coupling on surface ozone concentrations using a state of the art global Earth system model (CESM). The study builds on the recent informative work of Val Martin et al., GRL, 2014. A suite of sensitivity simulations is performed to unravel the relative roles of the modified biometeorological drivers in causing the altered surface ozone concentrations. The main overall conclusion is that surface ozone concentrations in the mid-latitude polluted temperate zone regions of Europe and North America are up to 6ppbv higher when the ozone-vegetation coupling is incorporated in the model framework. The sensitivity simulations indicate that the elevated surface ozone concentrations are mostly due to reduced dry deposition (stomatal closure on ozone damage) and increased isoprene emission (higher leaf temperatures due to reduced transpiration). Considerable effort and hard work has gone into conducting this challenging set of CESM simulations. The paper represents a valuable contribution to the literature in this emerging multidisciplinary field and deserves to be published in ACP once the following issues have been addressed. The findings may have important consequences in large-scale air quality modeling.*

*1. The main concern is the ozone damage function itself. The community accepts that flux-based damage schemes are more realistic than concentration schemes. However, it appears from the empirical parameters in Table 1 that in most cases (except photosynthesis for crops/grasses and stom. cond. for needleleaf) that the damage is independent of the CUO. Once the CUO has exceeded the threshold, the level of damage to photosynthesis and stom. cond. remains constant. This damage function does not seem realistic? Surely the level of plant physiological damage does depend (strongly!) on CUO. In their model, ozone vegetation damage for broadleaf biome is independent of CUO? More justification and explanation needs to be given as to the lack of dependence on CUO.*

As pointed out by the reviewer, several of the damage functions are independent of CUO. The damage functions were developed based on all available response data published through 2011 and constitute the most comprehensive database available for photosynthetic and stomatal responses to CUO to date (see Lombardozzi et al., 2013, Biogeosciences). There is so much variability in vegetation responses to ozone that often the photosynthetic and stomatal responses are not correlated with CUO. Even within one species across different studies that used the same experimental conditions, correlations are not evident (e.g., see Lombardozzi et al. 2013, Biogeosciences, Fig. 9, shown below).

[Figure]

**Fig. 9.** The correlation of photosynthesis (**a**) and stomatal conductance (**b**) to CUO from all studies that measured responses of wheat plants grown in pots exposed to 100 ppb $O_3$ in growth chambers and compared to control plants exposed to charcoal-filtered air. CUO is shown on a log scale, but linear analyses ($r^2$ and line equations) were performed on non-log transformed data. Different symbols represent different studies. Initial data considered (open circles and dashed line) were from a single study and demonstrated negative correlations between CUO and both photosynthesis ($r^2 = 0.59$, $p < 0.001$) and conductance ($r^2 = 0.19$, $p = 0.05$). Adding data from other studies measuring the responses of wheat plants to $O_3$ in similar environments (closed symbols) resulted in no overall correlation with CUO, though overall correlations exist in some individual studies (photosynthesis: triangles and dotted line, $r^2 = 0.15$, $p = 0.15$; squares and dot-dashed line, $r^2 = 0.61$, $p = 0.06$; conductance: triangles and dotted line, $r^2 = 0.28$, $p = 0.03$; squares and dot-dashed line, $r^2 = 0.004$, $p = 0.91$). The text in the figure and the solid regression line summarizes the information for all data points included in the figure. $r^2$ values are only included for overall significant correlations, and $n$ values are the number of data points included in the analyses.

Correlations of photosynthesis or stomatal conductance with CUO are often only evident within a given study that measures the response of the same plant through time, and there is no strong evidence that correlations of damage with CUO exist across studies in many instances. Using the results of individual studies as the basis for a global parameterization may not be a valid representation of global-scale processes, and we deem it more robust to use a function based on responses across multiple species, such as those developed by Lombardozzi et al. (2013), to be more in line with what the plant functional types represent in the model. In some cases, this means that the damage function is not correlated with CUO, as correlations with CUO across several species within a given plant functional type are not supported by data. The damage is applied after CUO reaches a certain threshold, so the calculation of CUO is still important in applying the damage functions. The same damage functions are tested in Lombardozzi et al. (2015). We lengthened the discussion about parametrization scheme as follows (p10 l367-378):

"Large variability in the responses of different plants to ozone leads to considerable uncertainties in any global-scale studies (Lombardozzi et al., 2013). **Such large variability in plant responses across different studies, in some cases, weakens the correlation between phytotoxic responses and CUO. Such correlation is usually more evident in individual studies, and in the parametrization schemes based on them (Sitch et al., 2007; Yue et al., 2014). The parameterization developed by Lombardozzi et al. (2013), based on the most comprehensive database available for photosynthetic and stomatal responses to CUO to date, is deemed more appropriate for the global scale of this study and the plant functional types represented in the model, despite the weaker correlation between plant responses and CUO as shown by the compilation of data across studies. The damage is applied after CUO reaches a certain threshold, so the calculation of CUO is still crucial to the application of the damage functions.** The model results could **possibly** be improved with more detailed plant-type-specific ozone damage parameterization, including better estimates of plant vulnerability to ozone that will help refine the ozone

uptake thresholds (Lombardozzi et al., 2015).**"**

*2. The results indicate that changes to photosynthesis have almost negligible impacts on the surface ozone concentrations. Isoprene emission is tightly coupled to photosynthesis (70-90%) that provides the energy and precursors for isoprene production. The MEGAN model does not include any direct connection to photosynthesis rate. Therefore, any influence of altered photosynthetic rate cannot change isoprene emission in this model framework. For example, ozone-induced photosynthetic reductions likely reduce isoprene emission and could potentially offset the temperature-related increases. Should a photosynthesis-based isoprene emission model have been used in this study to allow for all impacts on surface ozone? The current model misses this potentially important feedback. Are the surface ozone increases merely an artifact of the MEGAN model (or up to the 60% contribution)? Have the 6ppbv increases been largely overestimated?*

In this study, photosynthesis affects BVOC emissions indirectly through the long-term evolution of leaf area index (LAI) in response to ozone-induced damage on photosynthesis and carbon assimilation. LAI changes then affect BVOC emissions computed by MEGAN. As pointed out by the reviewer, the MEGAN model within CLM does not include direct, immediate connection between photosynthesis and BVOC emissions, and this could be a potentially important process offsetting some of the increases in BVOC as shown in Fig. 5(c). MEGAN is a very well established semi-empirical parameterization scheme for BVOC emissions, validated by a long line of observations and used ubiquitously in the community, and the dependence of BVOC emissions on photosynthetic rate is likely implicitly encapsulated (though not explicitly) in the activity factors for solar insolation, temperature, soil moisture and leaf age. The implementation of a new scheme for BVOC with a direct linkage to photosynthesis is beyond the scope of this study. We have lengthened the discussion about the limitations of MEGAN in the last section (p10 l386-392):

"…suggesting the need for further tuning of hydrometerological processes in the model. **Also, MEGAN does not consider the direct, immediate biochemical connection between photosynthesis and biogenic emissions, by which ozone damage on photosynthesis may directly reduce isoprene emission and partially offset the significant temperature-induced increase in isoprene emission as shown in Fig. 5c (Tiwari et al., 2015). Whereas the various environmental activity factors used in MEGAN to adjust baseline emissions may have implicitly encapsulated the biochemical connection with photosynthesis, further incorporating such connection into ozone-vegetation modeling warrants more in-depth investigation.** In general, we have the highest confidence in…"

*3. A reference that may be helpful: Tiwari et al., Ozone damage, detoxification and the role of isoprenoids - new impetus for integrated models, http://dx.doi.org/10.1071/FP15302*

Added and discussed as suggested (p10 l386-392); see our response above.

*4. Uncertainty ranges need to be provided e.g. on the 6 ppbv enhancements in surface ozone. These ranges could be based on 15-yr interannual variability from the multi-year simulations.*

6 ppbv of increase in surface ozone is the highest ozone enhancement simulated over Europe, hence the potentially largest impact of including ozone-vegetation coupling into CAM-Chem-CLM modeling framework. We have tested statistical significance of the surface ozone changes based on 15-yr summertime simulation results and revised Fig. 3 and Fig. 6(a) as suggested. See our responses to the first reviewer above.

*5. Figure 3 does not show substantial surface ozone feedbacks in China even though the Abstract (and text) claims up to 6 ppbv there.*

> We intend to highlight the regions where ozone-vegetation feedback is important, and "up to 6 ppbv" specifically refers to Europe among the three most affected regions (Europe, US, China). We revise the text now to make it clearer: **"… by up to 4-6 ppbv over Europe, North America and China"** in the abstract (p1 l18) and elsewhere (p9 l337).

*6. It is interesting that inclusion of the ozone-vegetation coupling worsens the evaluation of simulated ozone against AQ measurement networks: "this further highlights the urgency to revise other model processes and modules relevant for ozone simulations." What are the other model processes that need to be revised? How large are the uncertainties in the surface ozone feedback estimates provided here, given that the damage function is based on only 3 biome types, is independent of CUO and the isoprene emission scheme is independent of photosynthetic rate?*

> For instance, overestimation of stomatal resistance and related dry deposition scheme are demonstrated to be problematic and have a significant impact on surface ozone simulation (Val Martin et al., 2014). Lamarque et al. (2012) also mentioned issues related to emissions and resolution, and we further discussed these issues here:

> "…and other global and regional models (Lapina et al., 2014; Parrish et al., 2014). **Uncertain emissions, coarse resolution (Lamarque et al., 2012), misrepresentation of dry deposition process and overestimation of stomatal resistance (Val Martin et al., 2014) are all likely factors contributing to these high biases.** Inclusion of ozone-vegetation coupling…" (p7 l237-241)

> The uncertainty is now discussed at greater length here:

> "… from 18% to 22%, 31% to 35%, 14% to 22%, respectively. **Although there remains considerable uncertainty in the parameterization of ozone-vegetation coupling and in ozone simulations by Earth system models, we show that including ozone damage in a coupled climate-chemistry-biosphere framework can have a potentially significant impact on surface ozone simulations.**" (p7 l243-247).

> Issues relating to plant-CUO correlation and photosynthesis-isoprene linkages are also addressed (see above).

*7. Page 8, Line 298: "likely reflecting the relaxation of nitrogen limitation when photosynthesis is reduced". Needs more explanation in the text. What is "relaxation of nitrogen limitation"?*

> We now explain it at greater length:

> **"…** likely reflecting more carbon allocation to leaves to compensate the reduced photosynthetic rate **and relaxation of resource limitation as nutrients and water become less limiting upon lower photosynthetic and evaporative demands, as well as favorable hydrometeorological changes following reduced transpiration (e.g., enhanced vegetation temperature, precipitation and soil moisture)."** (p8 l310-314)

*8. Page 9, Line 334: "Transpiration rate is simulated to decrease by 6.4% globally, which is a larger change compared with the decrease estimated by Lombardozzi et al. (2015) and suggests an augmented effect due to the coupling between the atmosphere and ecosystems."*

*and discussed earlier in the paper. Why is the transpiration response 3 times larger in this work than in Lombardozzi et al., 2015? What exactly about the process of using coupled versus fixed ozone leads to this much larger impact?*

The changes found by Lombardozzi et al. (2015) are run with prescribed atmospheric climate forcings that are based on meteorological assimilation and therefore different from the simulated CAM atmospheric variables. The large difference in transpiration rates here compared with Lombardozzi et al. (2015) are likely due to both the differences between observed and simulated climate forcings, and feedbacks via changes in leaf area index, surface temperature and ozone concentration that further amplify the reduction in transpiration. We revise the statement as follows (p9 l349-353):

"Transpiration rate is simulated to decrease by 6.4% globally, which is a larger change compared with the decrease estimated by Lombardozzi et al. (2015), **who used prescribed instead of synchronously simulated atmospheric forcings. This also suggests an augmented effect on transpiration due to changes in foliage density, surface temperature and ozone concentration arising from ozone-vegetation feedbacks.**"

*9. Figure 2. I'm struggling to understand the CUO plot. Why is CUO relatively low in the mid-west U.S. crop belt, and e.g. very high in the UK where ozone concentrations are rather low. In some ways, CUO plot is inverse of surface concentration plot, which is understandable on the basis of the deposition sink. But, why are you even showing CUO when the ozone vegetation damage functions are independent of CUO (with minor exceptions discussed in point (1) above)?*

CUO accumulates over the growing season (defined as LAI > 0.5), and it depends highly on stomatal activities. Therefore, locations with high CUO are more likely to be those with either high surface ozone, high stomatal conductance or both, especially when they peak at the same time. That could explain much of spatial variation of CUO and its mismatch with ozone concentration. We now put the CUO plot in the supplementary material (Fig. S1) to avoid confusion.

*10. Page 8, Line 314: "which may have further ramifications for climate forcing because of the greenhouse effect of ozone." Increases in surface ozone have zero longwave radiative forcing because of the lack of thermal contrast with the surface. Ozone longwave radiative forcing is about changing ozone in the upper tropical troposphere.*

Removed as suggested.

*11. Figure 5(c). The only isoprene decreases are in tropical rainforest SE Asia regions. What causes these reductions localized to this region? Other tropical rainforest areas in S. America/Amazon and central Africa show increases.*

Decreases in isoprene emission in the rainforests in Southeast Asia are most likely dominated by the significant LAI reduction over the same region (Fig. 7e), where the highest ozone-induced reduction in LAI is simulated by the model. The Amazon and central Africa show increases or relatively small decreases in LAI that do not offset the temperature-induced increases in isoprene emission. Discussion related to LAI is revised as follows (p8 l307-309):

"… but the corresponding GPP and LAI changes are relatively small (~5% over regions concerned, **except for Southeast Asia, where the highest ozone-induced LAI reduction is simulated and leads to isoprene emission decrease despite higher**

**surface temperature**). …"

*12. Figure 6(b). The dry-deposition driven ozone changes plot shows random sporadic grid cells with very high positive and negative values. What is causing these very high responses in a few random grid boxes? Would it help to show responses that are only statistically significant with 95% C.I. only grid cells, or similar? Otherwise, the results are technically unconvincing.*

Revised as suggested. Fig. 6(b) is updated in the manuscript with top and bottom 5% extreme high and low values removed.

*Minor comments:*

*1. Abstract. Please remove "per se" and include quantitative description of statistical significance.*

(p1 l25) Removed "(which are significant per se)" as suggested. Statistical significance is now addressed at various places (see above).

*2. Page 5, Line 162. What is the time-step in the coupled land-atmosphere model?*

(p5 l164) 30 minutes. Added in the text now as suggested.

*3. Page 8, Line 310: "Many land surface modeling studies have estimated the direct effects of ozone on ecosystem production and land-atmosphere water exchange (Yue and Unger, 2014; Lombardozzi et al., 2015), and predicted a possible positive radiative forcing from the ozone-induced decline in the land-carbon sink (Sitch et al., 2007)." I suggest to change "Many" with "A few". 2-3 studies are not many.*

(p8 l326) Revised as suggested.

*4. Only use "significant" if you actually provide a quantitative statistical significance.*

We have tested the statistical significance (simple *t*-test, *p*-value = 0.10) of the differences in surface ozone concentrations using the 15-year time series of summertime mean data, and now mark the grid cells with significant changes in Fig. 3 and Fig. 6(a). See responses above.

*5. Need to include "+/-"in Figure 1 schematic e.g. for photosynthesis -> LAI as the model shows some increases in LAI when photosynthesis is reduced.*

The simulated increases in LAI in some regions have strong theoretical basis (see response to point #7 of reviewer #2). This is, however, not a universal result, and stands in contrast with results from many previous experimental studies that examined leaf responses to ozone exposure under carefully controlled experimental conditions. Also, such LAI increases are likely caused by hydrometeorological changes induced by transpiration changes, and do not represent a "direct" linkage between photosynthesis and LAI. We therefore prefer to leave the "+" sign as it is, and discuss our model results as we have done in the text:

**"…** likely reflecting more carbon allocation to leaves to compensate the reduced photosynthetic rate **and relaxation of resource limitation as nutrients and water become less limiting upon lower photosynthetic and evaporative demands, as well as favorable hydrometeorological changes following reduced transpiration (e.g., enhanced vegetation temperature, precipitation and soil moisture). These LAI**

**increases induced by ozone are not represented in Fig. 1 because they more likely reflect the fully coupled effect of changing hydrometeorology, instead of the direct effect of ozone on LAI as is typically observed in experimental studies (Ainsworth et al., 2012)."** (p8 l310-316)

Response to interactive comments by I. Super

*I would like to give credits to the work done in this paper. A relatively complete modeling framework for global ozone levels is presented, including important feedbacks related to ozone damage. This work would be a nice addition to existing literature. However, I would like to share two small comments.*

*On lines 88-92 the authors summarize several important feedbacks of ozone damage on the ozone concentration itself. One of the feedbacks is that a decrease in stomatal conductance increases the boundary layer height, therefore reducing the in-situ ozone concentration. I would like to add that another consequence is the increase of entrainment, which can have either a positive or negative impact on surface ozone concentrations.*

> We thank the commenter for the valuable comment, and have added the corresponding suggestion into our text:
>
> "… a cascade of meteorological changes**:** lower humidity that reduces the chemical loss rate of ozone; a thicker boundary layer that dilutes all pollutants**, but may enhance entrainment, which either increases or decreases surface ozone depending on the vertical ozone profile;** and higher temperature that enhances ozone mainly through increased biogenic emissions and higher abundance of $NO_x$ (Jacob and Winner, 2009). …" (p3 l90-94)

*Moreover, in the 'Model experiments' section the authors describe their different simulations. If I understand correctly, the PHT+COND run includes ozone impact factors for both the stomatal conductance and photosynthetic rate. Moreover, the photosynthetic rate affected by ozone uptake is used to calculate the stomatal conductance. So in principle there is a double impact on the stomatal conductance. In addition, the authors have chosen to do a simulation which includes an ozone impact factor for the photosynthetic rate, while calculating the stomatal conductance without such impact factor and with an intact photosynthetic rate. And vice versa for stomatal conductance (lines 184-188). I wonder why they use an intact photosynthetic rate as this simulation then has no physical meaning. I would rather suggest to use the photosynthetic rate corrected for ozone uptake, but setting the ozone impact factor for the stomatal conductance to 'no impact'. This would give a more realistic idea of the impact of the two separate pathways.*

> The photosynthetic rate affected by ozone is not used to calculate the stomatal conductance, hence double impact on stomatal conductance does not occur in our [PHT+COND] simulation. The parametrization of ozone damage on vegetation is based on Lombardozzi et al. (2012), in which author argued that ozone damage decouples photosynthetic rate and stomatal conductance, and should be applied independently in the numerical implementation. Therefore, we applied ozone damage to optimal photosynthetic rate and stomatal conductance values separately, avoiding double impact. We agree on the comment that [COND] and [PHT] simulations have no physical meanings with respect to what is actually happening within the plant cells. However, these two sets of sensitivity simulations give us tremendous insights on

which of the ozone-modified pathways is dominant in the feedbacks to surface ozone concentration.

**References**

Jacob, D. J., & Winner, D. A.: Effect of climate change on air quality, Atmos Environ, 43(1), 51-63, doi:10.1016/j.atmosenv.2008.09.051, 2009.

Lombardozzi, D., Sparks, J. P., Bonan, G., and Levis, S.: Ozone exposure causes a decoupling of conductance and photosynthesis: implications for the Ball-Berry stomatal conductance model, Oecologia, 169, 651-659, doi:10.1007/s00442-011-2242-3, 2012.

Lombardozzi, D., Sparks, J. P., and Bonan, G.: Integrating $O_3$ influences on terrestrial processes: photosynthetic and stomatal response data available for regional and global modeling, Biogeosciences, 10, 6815-6831, doi:10.5194/bg-10-6815-2013, 2013.

Sitch, S., Cox, P. M., Collins, W. J., and Huntingford, C.: Indirect radiative forcing of climate change through ozone effects on the land-carbon sink, Nature, 448, 791-U794, doi:10.1038/nature06059, 2007.

Tiwari, S., Grote, R., Churkina, G., Butlet, T.: Ozone damage, detoxification and the role of isoprenoids - new impetus for integrated models, Funct Plant Biol, 43(4) 324-336, 2016.

Yue, X., and Unger, N.: Ozone vegetation damage effects on gross primary productivity in the United States, Atmos Chem Phys, 14, 9137-9153, doi:10.5194/acp-14-9137-2014, 2014.

Val Martin, M., Heald, C. L., and Arnold, S. R.: Coupling dry deposition to vegetation phenology in the Community Earth System Model: Implications for the simulation of surface $O_3$. Geophys Res Lett, 41(8), 2988-2996, doi:10.1002/2014GL059651, 2014.

---

## Author Response (AR2)

Responses to co-editor on "Effects of ozone-vegetation coupling on surface ozone air quality via biogeochemical and meteorological feedbacks" by M. Sadiq, A. P. K. Tai, D. Lombardozzi and M. Val Martin. (MS No.: acp-2016-642)

Our point-by-point responses are provided below. Comments are *italicized*, our new/modified text is highlighted in **bold**, and highlighted in blue in the manuscript.

I have read in detail once more again the revised ms as well as the reviews, including the one public comment and your response to these two reviews and comment. Overall, it seems that you have tackled well the various raised comments and that the revisions you have included in the manuscript properly addresses some of the main criticism. My own criticism is that there are a number of occasions where the actual models response cannot be well explained leaving you to hypothesize what might be the explaining mechanisms behind that response, e.g., changes in chemistry over the EU, a stronger response in transpiration compared to the Lombardozzi et al. (2015) study, the reduced Nresource limitation. This is also further confirmed in your response to, especially some of the comments raised by reviewer #2, e.g., the link between CUO and surface ozone concentrations and the explanation for the different responses in isoprene in the various regions. It calls according to me for a more detailed analysis for specific sites, e.g., with 1-D model approaches of the similar system so that you can really nail down the various explanations for all these mechanisms. That would, however, be enough material for another paper and appreciate this paper providing a strong motivation to indeed further pursue such further in-depth process understanding studies also for those regions where these interactions and resulting feedbacks mechanisms seem to be most relevant. In this process of reading over the comments, your response and the resulting revisions, I still came across a couple of editors' comments that I would like you to address for my final decision.

You could consider to include the reference by Super et al., 2015JG002996, Cumulative ozone effect on canopy stomatal resistance and the impact on boundary layer dynamics and CO2 assimilation at the diurnal scale: A case study for grassland in the Netherlands, J. Geophys. Res. Biogeosci., http://dx.doi.org/10.1002/2015JG002996, after the statement on the impact of entrainment on ozone, lines 91, 92

(L91-92) Added as suggested:

"..., but may enhance entrainment, which either increases or decreases surface ozone depending on the vertical ozone profile (Super et al., 2015); and higher temperature that ..."

Line 179: "multiplied with.."

(L179) Revised.

Line 184: "Therefore, online ozone-vegetation coupling and feedback are included", I would propose here to state that the "previously discussed feedbacks are mostly included" also since you might not cover yet all the potential feedbacks (like ozone deposition impact on NOx exchange).

(L187) Revised as suggested.

Line 220: this finding that the ozone impact in the simulations with an interactive calculation of ozone concentrations and using prescribed ozone is an interesting one since it already suggests that the overall impact of the coupling is not that large, resulting in substantially differences in ozone, or is it more a coincidence dependent on what prescribed ozone climatology you used?

As pointed out by editor, overall impact of the coupling on CUO is not that large, even with substantially higher ozone level. This can be understood using Eq. (3), which shows that CUO depends both on surface ozone concentration and stomatal resistance, such that the effect of higher ozone concentration is partially counteracted by larger stomatal resistance, leading to proportionally smaller changes in CUO and the corresponding impacts on vegetation. The text is modified to clarify this point (L221-223):

"... leads to a similar pattern of ozone uptake by vegetation to the case using prescribed ozone due to the compensation between higher (lower) concentration and higher (lower) stomatal resistance, as reflected in Eq. (3)."

Line 230-232; Discussing your results by comparison with other global scale climate-chemistry studies; were you aware of the Ganzeveld et al. 2010 study on the impact of land cover and land use changes on  $O_3$  and atmospheric-chemistry climate interactions? I am mentioning this since in that study we showed that the impact of future land cover and land use changes on ozone was small also due to the role of compensating effects by inclusions of some of the same interactions as you consider in this study, e.g., changes in LAI (line 258), micro- and boundary layer meteorology and biogenic emissions, except of the ozone uptake impact.

This paper is now cited and discussed in the relevant section:

(L235-237) "This coupling effect is smaller than the potential ozone changes driven by anthropogenic emissions (up to +30 ppbv), but it more likely reflects compensation among various pathways (e.g., Ganzeveld et al., 2010). These simulated increases..."

(L343-345) "This coupling effect is significant in view of the 2000-2050 effects of climate and land cover changes on surface ozone (+1-10 ppbv) as found in previous work (Jacob and Winner, 2009; **Ganzeveld et al., 2010;** Tai et al., 2013), and should be considered in future air quality projection studies."

Line 280: "biogenic isoprene (or VOC?) emissions"; I think that you should stress that you consider only changes in biogenic emissions of isoprene and did not consider changes in biogenic N-emissions. By the way, so far you mainly referred to isoprene emissions but do changes in biogenic emissions also include changes in terpene and other BVOCs emissions? This is not completely clear. You should possibly stress this.

Revised accordingly in Sec. 4 (L286):

"A comparison between Fig. 6(a) and Fig. 3(a) shows that the changes in **biogenic VOC** emissions account for ~0-60% of the ozone increases over Europe, North America and China, ..."

Changes in isoprene emission, among all biogenic VOCs, are dominant in affecting tropospheric ozone concentrations. Isoprene emission alone comprises roughly half of the total emission of biogenic VOCs (Guenther et al., 2012) and is considered to be the most important biogenic hydrocarbon in atmospheric chemistry due to its highly reactive nature (L48-51). However, in our simulations, the embedded MEGAN module simulates emission fluxes of 19 different categories of biogenic compounds, including isoprene, monoterpenes and others. Although not dominantly, terpene reactions are linked to ozone. Therefore, turning off MEGAN and comparing between [PHT+COND\_nM] and CTR\_nM have indeed isolated the impact of total biogenic VOC emissions.

Line 287: "which provided the sensitivity of simulated ozone to perturbed dry deposition only without the complication of stomatal coupling with hydrometeorology". This modification doesn't read well. I would propose to rephrase this to: "which provides an approximated sensitivity of simulated ozone to perturbed dry deposition velocity only to separate this impact from that due to changes in hydrometeorology associated with changes in stomatal conductance, e.g., changes in mixing layer depth". This also links this statement better to the follow-up discussion/statement on mixed layer depth changes. I also think you should carefully refer to the term dry deposition being or the dry deposition velocity or dry deposition flux. Since you are referring in this part of the analysis to figure 5a, it should all read "dry deposition velocity".

Revised accordingly (L291-293, L301).

Line 291-292: "...feedback whereas over western Europe, where the lower chemical loss rate following reduced transpired water might have further enhanced the positive feedback"; see my suggestion for sentence change but also want to remark that such a statement about what might have caused a stronger response does not give a strong impression about the level of understanding of the actual explaining mechanisms. Are not there not additional diagnostics to indeed confirm that it is change in the chemical loss rate that explains the simulated response?

Revised as suggested (L296-298). As for additional diagnostics, unfortunately we did not archive the chemical loss rates as outputs in our 20-year simulations, but we were able to arrive at these explanations due to an examination of spatial patterns of changes in the associated hydrometeorological variables and an understanding of the interactive processes represented and not represented in CESM.

Line 314; here I read for the first time a reference to the role of soil moisture changes whereas I think this is an extremely essential component in this coupling mechanism. Has there indeed been hardly any change in soil moisture, has it not been focus of your study or was it already included in more detail in the other preceding studies? I am inquiring, also since I am informed that for example the soil moisture signal might be an essential component of the Mediterranean pollution response (the ecosystems being much less sensitive to ozone when stomatal closure due to reduced soil moisture is considered, see studies by Emberson et al. on the DOSE modelling and EMEP).

Changes in soil moisture and precipitation are shown below and also included in the supplement. We find increases in soil moisture (along with increases in precipitation) in many places where GPP and LAI also (counterintuitively) increase with ozone-vegetation coupling, hence we conclude that, by the understanding of model mechanisms, more favorable hydrometeorological conditions have contributed to this vegetation growth in these regions. This effect appears to dominate over the higher sensitivity of vegetation to ozone at higher soil moisture (as Emberson et al. suggested), which is also accounted for in the model though only weakly. Changes in soil moisture and precipitation tend to reinforce each other, however, leading to strong coupling both in the real world and in the model that is particularly difficult to isolate. We decide not to lengthen the discussion on this since soil moisture has not been our research focus, due to its less comprehensive parameterization in CLM and weaker linkage to photosynthetic processes (Bonan et al., 2014), and because as we have shown LAI changes have only a minor feedback effect on ozone. We now include the plots for soil moisture and precipitation changes in the supplement to clarify these points:

(L317-320) "... and relaxation of resource limitation as nutrients and water become less limiting upon lower photosynthetic and evaporative demands, as well as favorable hydrometeorological changes following ozone exposure (enhanced soil moisture and precipitation as shown in Fig. S5)."

(L400-402) "... hydrometeorological feedbacks introduce strong nonlinearity in the interactions between atmospheric chemistry, **soil moisture** and vegetation that is more difficult to isolate."

Figure S5. Percentage changes in (a) volumetric soil water content and (b) total precipitation rate in the [PHT+COND] case, where both photosynthetic rate and stomatal conductance are modified by ozone uptake, relative to the control case (CTR).

Regarding some of the revised figures, also the ones you included in the supplement; there is one striking one, Figure S3 which shows the relative changes in the  $O_3$  dry deposition velocity, indicating differences of -20% over Greenland. I am aware that there it is reflecting a relative difference for a very small absolute difference in Vd, which is mainly reflecting the snow-ice uptake rate and the turbulent and diffusive transport to the surface. You would guess that the snow-ice uptake would not change in your model (which is the case in most models) and that the change would reflect a small change in Vd in turbulent and diffusive transport. This is possibly also reflected in quite large differences in surface temperatures and heat fluxes expressed in Figure S4. Anyhow, so showing these relative differences including this "surprising" signal in such areas calls or for a better explanation of the mechanism, or you might prefer to only show the relative differences for those regions where this is some significant vegetation cover. This would serve to avoid getting too much distracted from the main point you want to make and requiring a more detailed explanation of such additional, complicated system responses.

Fig. S3 (a) and S4 (b,c) are updated accordingly by only showing differences where there is vegetation cover with annual mean LAI above 0.01, as shown below also: